Identification and validation of a prognostic model based on immune-related genes in ovarian carcinoma

Yu Min yumin@tjmuch.com 1 2 3
Li Dan 1
Zhang Li 1
Wang Ke kewang12@163.com 1 2 3
1 Department of Gynecologic Oncology, Tianjin Medical University Cancer Institute and Hospital , Tianjin , China
2 National Clinical Research Center for Cancer , Tianjin , China
3 Key Laboratory of Cancer Immunology and Biotherapy, Tianjin’s Clinical Research Center for Cancer , Tianjin , China
Marunaka Yoshinori
Electronic publication date: 2024 Oct 31
Publication date: 2024
Volume: 12
Electronic Location ID: e18235
Received 2024 Jun 11; Accepted 2024 Sep 15
Copyright: ©2024 Yu et al.
Copyright year: 2024
Copyright holder: Yu et al.
License: This is an open access article distributed under the terms of the Creative Commons Attribution License, which permits unrestricted use, distribution, reproduction and adaptation in any medium and for any purpose provided that it is properly attributed. For attribution, the original author(s), title, publication source (PeerJ) and either DOI or URL of the article must be cited.
License URL: https://creativecommons.org/licenses/by/4.0/

Keywords: Ovarian carcinoma, Immune, Prognostic model, GBP1P1, TCGA, ICGC

Funding: National Natural Science Foundation of China NO. 82272946 Tianjin Science and Technology Plan Project NO. 21JCYBJC01450 National Natural Science Foundation of China NO. 81602299 This work was supported by the National Natural Science Foundation of China (NO. 82272946), the Tianjin Science and Technology Plan Project (NO. 21JCYBJC01450), and the National Natural Science Foundation of China (NO. 81602299). The funders had no role in study design, data collection and analysis, decision to publish, or preparation of the manuscript.

==============================
Background

A novel valuable prognostic model has been developed on the basis of immune-related genes (IRGs), which could be used to estimate overall survival (OS) in ovarian cancer (OC) patients in The Cancer Genome Atlas (TCGA) dataset and the International Cancer Genome Consortium (ICGC) dataset.

Methods

This prognostic model was engineered by employing LASSO regression in training cohort (TCGA dataset). The corresponding growth predictive values of this model for individualized survival was evaluated using survival analysis, receiver operating characteristic curve (ROC curve), and risk curve analysis. Combined with clinical characteristics, a model risk score nomogram for OS was well built. Thereafter, depended on the model risk score, patients were divided into high and low risk subgroups. The survival difference between these subgroups was measured using Kaplan-Meier survival method. In addition, correlations containing pathway enrichment, treatment, immune cell infiltration and the prognostic model were also analyzed. We established the ovarian cancer cell line W038 for this study and identified the performances of GBP1P1 knockdown on a series of activities including cellular proliferation, apoptosis, migration, and invasion of W038 cells in vitro.

Results

We constructed a 25-genes prognostic model (TNFAIP8L3, PI3, TMEM181, GBP1P1 (LOC400759), STX18, KIF26B, MRPS11, CACNA1C, PACSIN3, GMPR, MANF, PYGB, SNRPA1, ST7L, ZBP1, BMPR1B-DT, STAC2, LINC02585, LYPD6, NSG1, ACOT13, FAM120B, LEFTY1, SULT1A2, FZD3). The areas under the curves (AUC) of 1, 2 and 3 years were 0.806, 0.773 and 0.762, in the TCGA cohort, respectively. Besides, the effectiveness of the model was verified using ICGC testing data. Univariate and multivariate Cox regression analysis exposes the risk score as an independent prognosis predictor for OS both in the TCGA and ICGC cohort. In summary, we utilized comprehensive bioinformatics analysis to build an effective prognostic gene model for OC patients. These bioinformatic results suggested that GBP1P1 could act as a novel biomarker for OC. GBP1P1 knockdown substantially inhibited the proliferation, migration, and invasion of W038 cells in vitro, and increased the percentage of apoptotic W038 cells.

Conclusions

The analyses of genetic status of patients with 25-genes model might improve the ability to predict the prognosis of patients with OC and help to select patients suit able to therapies. Immune-related gene GBP1P1 might serve as prognostic biomarker for OC.

Introduction

At present, ovarian cancer was the leading mortality of disease among gynecological cancers (Nash & Menon, 2020; Siegel et al., 2023). In 2022, 19,880 new OC cases and 12,810 OC deaths were projected to occur in the United States (Siegel et al., 2023). Ovarian cancer was a highly heterogeneous disease that 90% of OC were of an epithelial ovarian cancers (EOC) and nonepithelial ovarian cancers (NEOC) which accounted for approximately 10% of all OC comprised germ cell tumors (GCT) and sex cord-stromal tumors (SCST) (Saani et al., 2023). The morbidity of OC was showing a rising trend because of advances in diagnostic technology and a growing national awareness of their health (Jemal et al., 2011; Nash & Menon, 2020). The number of asymptomatic ovarian masses had increased with the use of prenatal ultrasonography and among ovarian tumors that complicated pregnancies, approximately 5% were malignant (Boussios et al., 2018). The 5-years survival of OC was poor owing to late diagnosis and frequent relapse (Gogineni et al., 2021; LaFargue et al., 2019). Currently, CA125 and HE4 were the only approved biomarkers for use in epithelial ovarian cancer; while multiple biomarkers on different platforms had been identified that might have potential as diagnostic tools for OC, they were still not sufficient for clinical practice (Ghose et al., 2024). Therefore, there was an urgently need to find effective systemic therapy and identify novel biomarkers of prognosis assessment for OC.

Immunotherapy had emerged as a practical and acceptable strategy for the treatment of advanced or metastatic cancers (Rocconi et al., 2021; Socinski et al., 2018). Unfortunately, the most widely studied immunotherapy, such as the anti-CTLA4, anti-PD1, or anti-PD-L1 monotherapy trials yielded only modest results in OC with an overall response rate (ORR) of 10–15% (Disis et al., 2019; Liu et al., 2019). However, the immune system seemed to play a role in ovarian cancer. This was reflected in the observation that tumor islets were infiltrated by T-cells in more than half of women with ovarian cancer (Zhang et al., 2003; Raspollini et al., 2005). Then, the researchers found that patients whose tumor was infiltrated by these T-cells have better clinical outcomes and the extent of infiltration was prognostic, not merely its presence or absence (Goode et al., 2017; Zhang et al., 2003; Raspollini et al., 2005). For the benefit of precision medicine and better prognosis in OC immunotherapy, it was critical to explore reliable immune-related biomarkers of prognosis and relevant molecular mechanisms.

To data, several previous studies provided compelling evidence of new immune-related genes (IRGs) for the prediction of cancer patient survival (Huo, Wu & Zang, 2020; Wen et al., 2020; Xia, Yan & Shen, 2021; Zeng et al., 2021; Zhang et al., 2021a; Zhang et al., 2021b). Those results also could explore a certain extent predictive value of IRGs for the survival of OC patients. However, it was still difficult to find a single effective and appropriate immune-related gene biomarker to identify and evaluate immune responses and to predict OS in OC patients.

Thus, the present study was aimed to develop and validate a prognostic model based on the IRGs and clinical features to estimate OS in OC patients; to evaluate its value of prognosis; to analyze correlations between treatment, immune cell infiltration and the prognostic model; to explore the potential mechanisms by gene enrichment analysis; and to conduct internal verification. We believed that our research constructed a reliable prognostic signature and hoped that this signature would be applied in clinical practice in the future.

Methods and Materials

Data extraction and preprocessing

UCSC Xena database (https://xena.ucsc.edu/) collected RNA-seq expression and clinical data from The Cancer Genome Atlas (TCGA) dataset and GTEx, including 379 tumor samples and 88 normal controls. In the present study, the TCGA ovarian cancer (OC) cohort and GTEx ovarian cohort were employed as the train cohort. The RNA-seq expression and clinical-pathological parameters of 85 OC patients were retrieved from International Cancer Genome Consortium (ICGC, https://dcc.icgc.org/). To operate the study data, the microarray probes with null gene test value were first eliminated and the remaining probes were mapped to human genes. The tumor samples absence of clinical information and survival data were excluded. Details of the steps were summarized in Fig. 1 through a flowchart.

Figure 1 Flow chart of the study design.

Screening for immune gene signature related to prognosis

The immune-related genes were downloaded from GSEA database (http://www.gsea-msigdb.org/gsea/index.jsp). The search keyword was “immune” and organism was “Homo sapiens”. A total of 458 immune-related genes were included in the subsequent analysis (Systematic name: M41715). Batch effects of the train dataset were removed using the “sva” package in R software. We then intersected all differentially expressed genes (DEGs) with immune genes to arrive at all of the immune DEGs in OC. Differentially expressed immune-related genes between normal and tumor tissues of the train dataset were identified using the Wilcoxon test (FDR < 0.05, |Log2(fold-change)| > 1). We additionally screened the immune DEGs related to prognosis through Univariate Cox Regression analysis. For the course of analysis, we used the R package “survival”. The screening criteria was p-value < 0.05. Furthermore, the immune gene correlation was used for the train cohort to predict integrative clusters and molecular subtype classifications through the “ConsensusClusterPlus” R package.

Description of prognosis-related molecular subtypes

Consensus clustering was conducted with the R software ConsensusClusterPlus package to distinguish the subgroups of OC according to those IRGs possessing of the highest variability. Initially, partial features and items were subsampled in the light of the data matrix by this algorithm, and all subsamples were classified as k groups based on the k-means. Their consensus values were calculated after repeating for the number specified by the user, and cluster stability was evaluated by using several clustering runs of algorithm. Subsequently, the proportion of cluster runs that divided into two items were regarded as the values of pairwise consensus, which were measured and saved within the consensus matrix for every k. Then, for every k, the 1-consensus value distance was applied to the eventual hierarchical agglomerative consensus clustering, which was reduced into k groups. Specially, the stability of clustering result was determined with the methods of the particular clustering approach into random data subsets, which was named as the “consensus” clustering (Hu, Yang & Sang, 2020; Wu et al., 2017). To generate more specific OC categories, more categories were required. Commonly, the consensus values set as 0 (white) and 1 (dark blue) were characterized with the color gradient. Furthermore, there were the same cluster items placed near each other in the matrix. As a result, the excellent consensus matrix exhibited the distinct color-coded heatmap. This heatmap consisted of blue blocks along the diagonal with the white background.

The construction of prognostic model

With the help of the “sva” package in R software, batch effects of the TCGA and ICGC dataset were deleted. Univariate Cox regression analysis for screening the prognostic genes with the “survival” R package was employed to constitute a reliable prognosis signature. In order to make a feature selection and screen the importance of prognostic-related genes with p < 0.05, the classic random forest algorithm was utilized. Thereafter, based on the parameters of the least absolute shrinkage and selection operator (LASSO) COX regression analysis, we build the prognostic signature with 1,000-fold cross-validation with the “glmnet” package in R software. Then the risk score of every patients was determined according to the following formula = ∑Coefgene × Expgene, where Coefgene denotes the each prognostic gene coefficient, the Expgene denotes the expression of each gene (Dong & Xu, 2019). After obtaining the median risk score, all patients were divided into high-risk and low-risk groups.

Evaluation of prognostic model

The time-dependent receiver operating characteristic (Rocconi et al., 2021) analysis with the “timeROC” package in R software was chosen to evaluate the predictive accuracy of the signature. Furthermore, the risk curve was created by the “pheatmap” R package to evaluate the prognostic model. In order to test the feasibility of prognostic model as an independent predictor, the indicators containing clinical traits and risk values in train cohort and test cohort was measured using univariate and multivariate Cox regression analyses.

Analysis for GO and KEGG enrichment pathway

GO analysis including cellular component, molecular function, and biological process to illustrate the multiple cellular functions of certain genes.

KEGG is an analysis method for uncovering the specific biological pathways of certain enriched genes. Therefore, GO and KEGG pathway analysis were conducted with the “clusterProfiler”, “org.Hs.eg.db”, “enrichplot”, and “ggplot2” R package according to DEGs with high-risk and low-risk groups of OC patients. The significant pathways were enriched with q-value < 0.05 (the corrected p-value).

Drug sensitivity analysis

In order to examine the drug sensitivity and tolerance effects of genes involved in developing the prognostic model, the transcriptome and FDA-certified drug sensitivity related data were downloaded from the CellMiner database (https://discover.nci.nih.gov/cellminer/). The Pearson correlation analysis was utilized to demonstrate the relationships between gene expression and drug sensitivity.

Estimation of immune cell infiltration

To examine the connection between the prognostic genes and immune cell infiltration, we analyzed TCGA cohort using online tool of GEPIA (http://gepia.cancer-pku.cn/) and TIMER (https://cistrome.shinyapps.io/timer/). Median expression value of prognostic genes was set as critical point, and divided into high and low expression groups. To further evaluate immune cell infiltration, the single sample gene set enrichment analysis (ssGSEA) was selected to study the different infiltration degrees of immune cell subgroups between above two high and low expression groups using the R package “GSVA”.

Single cell sequencing data download and processing

The scRNA-seq dataset GSE118828 was downloaded from the GEO database (http://www.ncbi.nlm.nih.gov/geo/). Total number of 16 OV samples were included in the cohort in addition to benign and normal tissues. The R package Seurat (version 4.3.4) (Butler et al., 2018) was used to conduct the process of quality control (QC). The data filtering parameters were set as follows: 1. Genes expressed in at least three single cells; 2. RNA in cells was greater than 50; 3. Mitochondrial genes less than 5%. At the bottom of QC, the cell populations were tagged based on marker genes. These markers were recognized as major immune cell types and visualized with the dimensionality reduction algorithm uniform manifold approximation and projection (UMAP).

Establishment of specific ovarian cancer cell line

Tissue sample of primary OC was obtained from Tianjin Medical University Cancer Institute and Hospital after approval of the Institutional Ethics Committee (NO. bc2023167) and informed written consent from the patient’s family. This clinical specimen was donated from a 61-year-old Chinese female patient with advanced ovarian carcinoma. The willingness to provide verbal informed consent was abstained. Tumor specimen was washed with sterile phosphate-buffered saline (PBS) for three times, and then re-suspended with RPMI-1640 medium containing 10% FBS, 5–10% cell-free ascites, penicillin G (100 U/ml), streptomycin (100 µg/ml). Subsequently, it was transferred into a 10 cm petri dish at 37 °C and 5% CO2. After the confluence rate up to of 80–90%, these primary cultured cells were sub-cultured at a ratio of 1:1. This cell line was considered as a continuous cell line after 50 passages and named W038. The detail morphological properties of W038 were monitored with the help of an inverted microscope. Meanwhile, cells treated with hematoxylin and eosins (HE) were detected using a light microscope.

Transfection

The GV298 lentiviral particles consisting of GBP1P1 shRNA sequence (5′-GCCAAGTCTG GTCACTAAACT-3′) were designed and synthesized by Jikai Kiin Technology Co. LTD (Shanghai, China). Lentiviral particles containing 5 µg/mL polybrene were added into W038 cells. After the transfection for 48 h, targeting cells were selected with 2 µg/mL of puromycin for two months. Then the resistant colonies were pooled, expanded, and named sh-GBP1P1 W038. The negative control cells with disordered shRNA (5′-TTCTCCGAACGTGTCACGT-3′) were named sh-CTRL W038. The cells of GBP1P1 knockdown were identified by quantitative real-time polymerase chain reaction (qRT-PCR), and fluorescence imaging using fluorescence microscopy (Lecia, Wetzlar, Germany). The minimum information for publication of quantitative real-time PCR experiments (MIQE) was listed in the Supplemental Material.

Immunohistochemistry staining

Immunohistochemical studies were performed on paraffin section with precipitation from cell centrifugation. The sections (4 µm) were heated at 60 °C for 1 h, deparaffinized in xylene, and rehydrated with graded ethanol. Then samples were heated in citrate buffer (pH 6.0) for 2 min to retrieve antigens. Endogenous peroxidase activity was quenched using a methanol and hydrogen peroxide bath for 20 min. After pretreatment, slides were loaded with the primary antibody (1:200) overnight at 4 °C. Next, the biotinylated secondary antibody named goat anti-human IgG (Santa Cruz Biotechnology, USA) was labeled with streptavidin–horseradish peroxidase (HRP) using a DAB staining kit (Maixin Biotechnology, China). Negative controls were performed by omission of primary antibody. Positive cells showed brownish yellow in the cytoplasm or the cell membrane. These images were acquired with an Olympus BX51 microscope. Five high-power fields were chosen randomly for histological evaluation. The 2 pathologists independently reviewed the antibody stained slides along with the HE slides 1 to 3 months later.

The assays of cell proliferation, invasion, and migration

The CCK-8 kit was carried out with a density of 2,000 cell/well in 96-well plates. Cells were then treated with 10 µL of CCK-8 solution (Dojindo Molecular Technologies, Inc., Rockville, MD, USA) for 60 min. Microplate spectrophotometer was used to determine the optical density at 450 nm of cells at different time. The invasion assay was performed as follows: 4,000 cells suspended in 200 µL of serum-free RPMI 1640 media were seeded into the upper chambers, which were coated with matrigel (1:5 diluted solution; BD Biosciences); then dried at 37 °C for 2 h; 650 µL RPMI 1640 media with 10% FBS was next added into the lower chambers. After the incubation for 48 h, the filtered cells were fixed in 4% paraformaldehyde and loaded with 1% crystal violet at 37 °C and 5% CO2; the stained cells from five selected randomly views were counted with a microscope at a 200× magnification. For the migration assay, the ibidi Culture-Insert system was used according to the manufacturer’s in structions (ibidi, Martinsried, Germany).

Flow cytometry apoptosis assay

After addition with 3µg/mL carboplatin for 24 h, trypsin was used to digest OC cells which then washed with PBS for twice. Cells were suspended with 1× binding buffer at the density of 1 × 106 cells/mL in flow tubes, and volume of cell solution was 100 µL. Thereafter Annexin V-IF488 (5 µL) and 7-AAD (5 µL) solutions were added into each tube. After the reaction for 15 min, another 400 µL of 1× binding buffer was added to stop the reaction. Eventually, Flow cytometry was used for on-board detection within 1 h. The percentage of apoptotic cells in each group was evaluated by calculating the sum of Annexin V + 7-AAD + and Annexin V + 7-AAD − cells. The flow cytometry files could be found here: https://doi.org/10.6084/m9.figshare.25991899.v1.

Statistical analysis

R version 4.3.4 was used to conduct the statistical analysis of this research. A difference of p value < 0.05 indicated statistical significance.

Results

Determining the consensus clustering result

According to the expression profile of IRGs, we employed a consensus clustering algorithm to categorize the TCGA OC patients into de novo groups. In comparison with traditional k-mean and hierarchical clustering algorithms, consensus clustering was illustrated to be more robust and insensitive to random starts, and had been broadly used to recognize biologically meaningful clusters (Monti et al., 2003). A total of 243 differentially expressed IRGs (TCGA tumor vs GETx normal tissues, FDR < 0.05, |Log2 (fold-change) | > 1) were used for consensus clustering to identify selection of adequate groups. As indicated in Fig. 2A and Fig. S1, the color-coded heat-map showed that TCGA OC samples were divided into 2 IRGs subgroups performed better than those clustered into more than two categories based on consensus clustering. The color gradients represented consensus values from 0 to 1 (white corresponded to 0 and dark blue to 1). To sum up, the dataset was finally partitioned into two subgroups: cluster 1 (N = 180) and cluster 2 (N = 155). As implied by results of Kaplan–Meier survival analysis, differences in the prognosis were statistically significant across those two de novo clusters (Fig. 2B, P < 0.001). Additionally, the 243 IRGs expression heat-map of all OC patients in the training cohort were shown in Fig. 2C. We could see from heat-map that there was no correlation between two subgroups and clinical characteristics such as age, stage, grade, tumor residue and race.

Figure 2 Consensus clustering for the immune related genes in the TCGA dataset.

(A) Color-coded heatmap corresponding to the consensus matrix for k = 2 obtained by applying consensus clustering. Color gradients represent consensus values from 0 to 1; white corresponds to 0 and dark blue to 1. (B) Survival curves for each immune-related subtype in the TCGA database. The horizontal axis represents survival time (years), and the vertical axis represents the probability of survival. (C) Heatmap of the 243 IRGs expression profiles.

Figure 3 Construction and evaluation of the prognostic model based on IRGs.

(A) LASSO regression of the 20 IRGs. (B) Cross-validation for optimizing the parameter in LASSO regression. (C) Kaplan-Meier curve showing the survival difference between low-risk and high-risk groups in TCGA OC patients. (D) ROC curve of the risk score in predicting 1-, 2-, and 3-years OS in TCGA OC patients. (E) Risk curves showing the distribution of risk score in TCGA OC patients. (F) Risk curves showing the distribution of survival status of TCGA OC patients.

Construction and assessment of the prognostic model based on IRGs

In order to establish a reliable prognosis signature, batch effects of the TCGA, GETx and ICGC dataset were removed using the “sva” package in R software. Here, we assessed the prognostic value of the 243 differentially expressed IRGs in OC patients using univariate Cox regression analysis (Tables S1 and S2). Afterwards, 25 genes were taken to establish a prognostic risk model for categorizing OC patients into low-risk and high-risk group with discrete OS by LASSO Cox regression analysis. As indicated in Figs. 3A, 3B and S2, the 25 genes were appropriate for building the prognostic risk model for OC patients depending on the primary screen of the IRGs. The risk score was calculated as follows: risk score = 0.0361 × TNFAIP8L3 + 0.0062 × PI3 + 0.0053 × TMEM181 + (− 0.1056 × GBP1P1) + (− 0.0018 × STX18) + 0.0093 × KIF26B + (− 0.0066 × MRPS11) + 0.0135 × CACNA1C + (− 0.0124 × PACSIN3) + (− 0.0027 × GMPR) + (− 0.0040 × MANF) + 0.0003 × PYGB + (− 0.0008 × SNRPA1) + (− 0.0103 × ST7L) + (− 0.0004 × ZBP1) + (− 0.0114 × BMPR1B-DT) + 0.0030 × STAC2 + (− 0.0247 × LINC02585) + (− 0.0252 × LYPD6) + (− 0.0064 × NSG1) + (− 0.0059 × ACOT13) + 0.0037 × FAM120B + (− 0.0061 × LEFTY1) + (− 0.0513 × SULT1A2) + (− 0.0145 × FZD3). Furthermore, all the OC patients were divided into two groups based on the median risk scores calculated by the above formula. Patients in the high-risk group had significantly shorter survival time than the patients in the low-risk group via Kaplan–Meier survival analysis (p < 0.001, Fig. 3C). The ROC curve suggested the predictive ability of the risk score with regard to prognosis; the areas under the curves (AUC) of 1, 2 and 3 years were 0.806, 0.773 and 0.762, respectively (Fig. 3D). Subsequently, the risk curves were used to evaluate the prognostic models by setting the median risk score as the cut-off point (Figs. 3E and 3F). As the risk score rose, the death numbers increased, and the survival time decreased. Univariate and multivariate Cox regression analysis revealed the risk score as an independent prognosis predictor for OS in TCGA cohort (Table 1).

Validation of the prognostic model in ICGC cohort

A group of 85 ICGC OC patients with complete survival information served as an external validation cohort. The formula mentioned above was used to calculate the risk score, and patients were divided into two groups with the same cutoff. As indicated in Fig. 4A, the high-risk group in ICGC cohort also showed poor outcomes as demonstrated by Kaplan–Meier survival analysis (p < 0.05). The AUC obtained from the ROC curve for OS at 1, 2, and 3 years were 0.796, 0.743, and 0.716, respectively (Fig. 4B). Similar to the TCGA cohort, the risk curves were used to show the distribution of risk score and survival status. We could see from Figs. 4C and 4D that the death numbers increased and the survival time decreased as the risk score rose. Univariate and multivariate Cox regression analysis exposes the risk score as an independent prognosis predictor for OS in ICGC cohort (Table 1). These results further confirmed the hardiness of the signature in predicting overall survival of OC.

Table 1 Univariate and multivariate cox regression analyses in OC.

Cohort	Variables	Univariate analysis	Multivariate analysis	
		P-value	HR	95% CI	P-value	HR	95% CI	
TCGA	Age	<0.001	1.026	1.013–1.040	0.006	1.019	1.006–1.033	
Stage	0.017	1.473	1.071–2.026	0.094	1.327	0.953–1.849	
Grade	0.284	1.279	0.815–2.008				
Riskscore	<0.001	56.287	27.808–113.935	<0.001	54.137	26.167–112.005	
ICGC	Status	0.004	4.591	1.630–12.927	0.003	4.856	1.695–13.908	
Age	0.327	1.016	0.985–1.047				
Stage	0.077	0.492	0.224–1.080				
Riskscore	<0.001	3.649	1.834–7.262	<0.001	3.607	1.880–6.919	

Figure 4 Validation of the prognostic model based on IRGs in ICGC OC patients.

(A) Kaplan-Meier curve showing the survival difference between low-risk and high-risk groups. (B) ROC curve of the risk score in predicting 1-, 2-, and 3-years OS. (C) Risk curves showing the distribution of risk score. (D) Risk curves showing the distribution of survival status.

GO and KEGG pathway enrichment analysis of the different risk groups

To investigate the potential function and mechanisms of the low-risk and high-risk groups, we summarized differential genes in two risk groups to conduct GO and KEGG enrichment analysis in TCGA OC cohort. First, the heat-map of the 25 genes involved in the construction of the prognostic model in the different risk group was shown in Fig. 5A. We could see from the heat-map that the high-risk genes were TNFAIP8L3, PI3, TMEM181, KIF26B, CACNA1C, PYGB, STAC2 and FAM120B.

Figure 5 GO and KEGG pathway enrichment analysis in TCGA OC patients.

(A) Heatmap of the 25 genes involved in building prognostic models in the high-risk or low-risk group. (B) GO analysis was performed to detect biological processes. P < 0.05. (C) KEGG pathway analysis results indicated the enriched signaling pathways. P < 0.05.

The top five GO enrichment biological processes were oxidative phosphorylation, mitochondrial gene expression, mitochondrial translation, translational elongation and mitochondrial respiratory chain complex assembly (Fig. 5B, q-value < 0.05). Additionally, KEGG analysis revealed that several immune-related pathways were enriched such as antigen processing and presentation, human T-cell leukemia virus 1 infection, and leukocyte transendothelial migration (Fig. 5C, q-value < 0.05).

Drug sensitivity analysis

CellMiner (http://discover.nci.nih.gov/cellminer) was web-based applications for mining publicly available genomic, molecular, and pharmacological datasets of human cancer cell lines (Reinhold et al., 2012; Reinhold et al., 2015). We obtained the top 16 drugs approved by the US Food and Drug Administration (FDA) with the most statistically significant differences, by performing a separate drug sensitivity analysis on genes involved in the construction of the prognostic model (Fig. 6). The results showed that the expression of NSG1 was positively correlated with the sensitivity of vemurafenib and dabrafenib. The expression of ZBP1 was positively correlated with the sensitivity of LDK-378 and alectinib. The expression of GBP1P1 was positively correlated with the sensitivity of idelalisib and IPI-145. The expression of KIF26B, STX18 and CACNA1C were positively correlated with the sensitivity of zoledronate, nelarabine and idelalisib, respectively. It was indicating that the higher the expression of NSG1, ZBP1, GBP1P1, KIF26B, STX18 and CACNA1C, the stronger the sensitivity to the abovementioned drugs. The expression of GMPR was positively correlated with the sensitivity of vemurafenib and dabrafenib, but it was negatively correlated with the sensitivity of brigatinib. The expression of PYGB was positively correlated with the sensitivity of vemurafenib, but it was negatively correlated with the sensitivity of oxaliplatin. In addition, the higher the expression of PACSIN3 in OC patients, the patient’ s drug resistance to carmustine and bendamustine was stronger. We also predicted the correlation between drug sensitivity of targeted drugs and commonly used chemotherapy agents and patterns of 25 IRGs expression involved in the construction of prognostic models for ovarian cancer (Fig. 6 and Fig. S3). We found that targeted drugs were more significantly associated with gene expression patterns than conventional chemotherapy drugs.

Figure 6 Gene-drug sensitivity analysis based on the CellMiner database.

GBP1P1 (LOC400759) was involved in immune cell infiltration in OC

To find biological function for genes involved in building prognostic models, we analyzed the correlation between gene expression and abundance of immune infiltrates by online tool of GEPIA and TIMER. Among all the 25 prognostic genes, the result showed that significant correlation between GBP1P1 expression and immune cell infiltration was the most robust (Fig. 7A and Figs. S4–S7). These findings suggested that GBP1P1 might be involved in the immune cell infiltration in OC.

Figure 7 Expression of GBP1P1 in OC.

(A) Correlation analysis of GBP1P1 and immune cell infiltration in TCGA cohort. (B) Single-cell RNA-sequencing analysis to explore the cell localization of GBP1P1 gene in GSE118828 cohort. The cell types annotated by marker genes. (C) Expression of GBP1P1 in five cell types.

Cell Localization of GBP1P1

Based on scRNA-seq data of GSE118828, we obtained gene expression profiles from 16 OV samples in different cell types. We accomplished PCA using the top 1,500 variable genes to reduce the dimensionality, and 10 cell clusters were identified. Thereafter, the cell identity of each cluster was annotated using a reference dataset from the Human Primary Cell Atlas, and finally determined five cell types. T cells, endothelial cells, epithelial cells, smooth muscle cells and monocyte, respectively (Fig. 7B). Then, GBP1P1 was distributed on all five cell types, with the highest distribution on epithelial cells (red) as shown in Fig. 7C.

GBP1P1 knockdown inhibits malignant phenotype of ovarian cancer in vitro

We established the ovarian cancer cell line W038 for this study. To evaluate the specific role of GBP1P1 in ovarian cancer, we examined the effects of GBP1P1 knockdown on the proliferation, apoptosis, migration, and invasion of W038 cells in vitro. The W038 cell line was derived from ascites from patients with advanced ovarian cancer. After 50 passages, the cell morphology was homogeneous and cells with mesenchymal morphology were not observed (Fig. 8A). H&E staining showed that W038 cells were adenocarcinoma cells. Immunocytotochemical study revealed that W038 cell line was positive for cytokeratin (CK7), P53 and pair box gene8 protein (PAX-8), but negative for Wilms tumor protein (WT-1), estrogen receptors (ER), progesterone receptors, caudal type homeobox 2 (CDX-2) (Fig. 8B). To determine the efficiency of siRNA depletion, fluorescence images and qRT-PCR were performed. As shown in Fig. 8C, si-GBP1P1 and red fluorescent protein were co-expressed on the carrier, and the transfection efficiency was directly visualized by fluorescence, indicating that siRNA had been transferred into the W038 cell. qRT-PCR analysis for viral transcripts showed GBP1P1 was significantly reduced in sh-GBP1P1 W038 cells compared with sh-CTRL W038 cells (Fig. 8D). The results of CCK8 assays specified that GBP1P1 knockdown significantly inhibited the proliferation rate of W038 cells (Fig. 8E). The results of transwell assay demonstrated that the W038 cells exhibited markedly decreased invasion upon GBP1P1 knockdown (Figs. 8F and 8G). The results of wound healing assay revealed that the knockdown of GBP1P1 significantly decreased the migration of W038 cells (Figs. 8H and 8I). The apoptosis was observed by flow cytometry (IF488-Annexin V/7-AAD) and the results showed that the apoptosis rate of W038 cells increased significantly in the sh-GBP1P1 groups (Figs. 8J and 8K).

Figure 8 Knockdown of GBP1P1 inhibited proliferation, migration and invasion of ovarian cancer cells.

(A) Morphology of W038 cell line. (B) W038 cells were verified through HE and immunohistological images of CK7, P53, PAX-8, WT-1, ER, PR, CDX-2 antibody staining of tumor slides. (C) The transfection efficiency of sh-GBP1P1 in the W038 cell line detected by fluorescence imaging (cells with GV298 lentiviral particles transfected; red). (D)The transfection efficiency of sh-GBP1P1 in the W038 cell line detected by q-PCR. (E) The CCK-8 assay was used to detect the effect of GBP1P1 knockdown on the proliferation of W038 cell line. (F) Representative images of the invasion assay. (G) Statistical analysis of the invasion assay results after knockdown of GBP1P1 in the W038 cell line. (H) Representative images of the wound healing assay. (I) Statistical analysis of the wound healing assay results after knockdown of GBP1P1. (J) Cell apoptosis in each group was detected by flow cytometry. (K) Statistical analysis of the cell appoptosis results after knockdown of GBP1P1.

Discussion

Immunotherapy had become a promising clinical treatment strategy for several refractory carcinomas, however its roles in OC was limited (Bellone et al., 2018; Mesnage et al., 2016; Mittal et al., 2014). Of the solid tumors reviewed, breast and ovarian cancers have proved efficacy in the combination of Poly (ADP-ribose) polymerase (PARP) inhibition and immunotherapy, predominately in BRCA-mutated tumors (Aliyuda et al., 2023; Setordzi et al., 2021). PARP-based therapies operate through the inhibition of single-strand DNA repair resulting in DNA damage, growth tumor mutational burden, making the tumor a more amazing target for immunotherapy (Aliyuda et al., 2023; Vikas et al., 2020). The challenges majored in recognition of the appropriate patients and finding procuctive combination therapeutic targets to magnify its clinical efficacy. Freshly, prognostic models according to some disease-related genes or other biomarkers for improving the prognosis of carcinomas had attracted considerable attention (Li et al., 2021; Wang et al., 2020; Yuan et al., 2021).

We regrouped TCGA OC patients into two subgroups according to the expression of 243 differentially expressed IRGs. Then, Lasso penalized Cox regression analysis was accomplished to construct a new 25-genes prognostic model based on the above two subgroups. We assessed the capability of the 25-genes prognostic model in the training dataset and also in the testing dataset via ROC curve analysis, risk curve analysis, univariate and multivariate Cox regression analyses. The AUCs of the prognostic model for predicting the 1-, 2-, and 3-year OS of training dataset were 0.806, 0.773 and 0.762, respectively, and 0.796, 0.743, and 0.716, respectively for the testing dataset. These results also showed that the 25-genes prognostic model had an unusual capability for the prediction of survival and could be a durable and productive model of prognosis. In addition, KEGG analysis indicated that several immune-related pathways were enriched (Fig. 5). Moreover, the prognostic model had other effects on clinical applications, for example using these genes to find potential therapeutic targets and drugs, which may provide new ideas for the diagnosis and treatment of OC (Fig. 6). If it could be combined with patient-derived organoids preclinical models to predict clinical outcomes, the prediction results would be more accurate (Vlachogiannis et al., 2018; Yao et al., 2020).

Six genes (TMEM181, MRPS11, PACSIN3, NSG1, ACOT13, STX18) had never been reported to be associated with tumor prognosis, which came from the 25 genes involved in the construction of OC prognostic models. The remaining 19 genes (TNFAIP8L3, PI3, GBP1P1 (LOC400759), KIF26B, CACNA1C, GMPR, MANF, PYGB, SNRPA1, ST7L, ZBP1, BMPR1B-DT, STAC2, LINC02585, LYPD6, FAM120B, LEFTY1, SULT1A2, FZD3) were reported to be associated with tumor prognosis, either alone or in combination with other genes (Angius et al., 2019; Caba et al., 2016; Cai et al., 2021; Chang & Dong, 2021; Chen et al., 2022; Kim et al., 2016; Liang et al., 2022; Lin et al., 2021; Liu et al., 2022; Padmavathi et al., 2018; Riker et al., 2008; Siamakpour-Reihani et al., 2015; Sui et al., 2018; Yang et al., 2019; Zhang et al., 2021a; Zhang et al., 2021b; Zhao et al., 2020; Zhao et al., 2022; Zheng et al., 2021). To date, GBP1P1 was found to be as significantly correlated with overall survival (OS) only in hepatocellular carcinoma (HCC) and OC patients. In this study, according to the results of GBP1P1 involvement in the immune cell infiltration, GBP1P1 was expected to become a novel target for immunotherapy in OC in the futrure. So, single-cell sequencing technology was used to reveal the changes in the immune microenvironment of OV. GSE118828 revealed the heterogeneity of OV and the gene GBP1P1 is up-regulated on T cells (red) as showed in Fig. 7. We could notice that GBP1P1 had a positive impact on prognosis and cell phenotype verified tests suggested that GBP1P1 silencing could inhibit cell proliferation, migration, invasion and promote apoptosis by constructing a GBP1P1 knockdown model in W038 cells. Therefore, we believed that GBP1P1 was associated with the prognosis of OC, which was determined by the combined influence of complex tumor microenvironment. The analysis of the results indicated that GBP1P1 as a diagnostic gene could provide a new treatment strategy for OC in combination with immunotherapy.

Limitations existed in this study. First, our study was not generated immune subtyping for OC based on IRGs subgroups. Although the data of the research was mainly based on the TCGA, GTEx ,GEO and ICGC datasets, the patients in our own cohort need to be tested and expanded. Moreover, if we could find fewer genes involved in building prognostic models, clinical applications will be much better. Additionally, the potential molecular mechanisms of GBP1P1 in OC must be elucidated in future studies.

Conclusion

The 25-genes prognostic model was constructed and assessed. It was verified to be effective in the testing data. Our findings provided a theoretical basis for developing prognostic model constructed by IRGs clustering subgroups, predicting patient prognosis and selecting patients for combination therapeutic drugs.

Supplemental Information

Supplemental Information 1 MIQE checklist

Supplemental Information 2 Supplementary material

Additional Information and Declarations

Competing Interests

Author Contributions

Human Ethics

Data Availability

The authors declare there are no competing interests.

Min Yu conceived and designed the experiments, performed the experiments, analyzed the data, prepared figures and/or tables, authored or reviewed drafts of the article, and approved the final draft.

Dan Li performed the experiments, analyzed the data, authored or reviewed drafts of the article, and approved the final draft.

Li Zhang analyzed the data, prepared figures and/or tables, and approved the final draft.

Ke Wang analyzed the data, authored or reviewed drafts of the article, and approved the final draft.

The following information was supplied relating to ethical approvals (i.e., approving body and any reference numbers):

The study was approved by the Ethics Committee of Tianjin Medical University Cancer Institute and Hospital (NO. bc2023167).

The following information was supplied regarding data availability:

Publicly available datasets were analyzed in this study and the raw data is available from the GTEx, TCGA and ICGC datasets.

The data is available at NCBI GEO: GSE118828.

The codes and intermediate files are available at figshare: Yu, Min; Li, Dan; Zhang, Li; Wang, Ke (2022). codes and intermediate files. figshare. Dataset. https://doi.org/10.6084/m9.figshare.19583854.v1.

The flow cytometry files are available at figshare: Yu, Min; Li, Dan; Zhang, Li; Wang, Ke (2024). Flow cytometry files. figshare. Figure. https://doi.org/10.6084/m9.figshare.25991899.v1.

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
