# Peer review of "Identification and validation of a prognostic model based on immune-related genes in ovarian carcinoma"

_PeerJ, doi:10.7717/peerj.18235_

## Round 0.1 · original submission · Major Revisions

Dear Dr. Wang,

If you feel you can revise your manuscript according to the reviewers' comments, please revise your manuscript and submit it. Please also send us your written responses to each of the reviewers' comments.

Yours,

Yoshi

Prof. Yoshinori Marunaka, M.D., Ph.D.

Reviewer 1 ·

Basic reporting

Wang et al presented a well written manuscript providing sufficient background setting the stage for the scientific question the team is trying to explore.

Experimental design

Wang et al established a 25 immune gene based prognostic signature for ovarian cancer. They further explore the biological relevance of the signature obtained. They also identified a novel biomarker.

Suggested improvement:
In fig 1 it’s not very clear how DEGs were obtains, add specific comparison that was made to obtain DEGs before comparing them with the immune genes

Validity of the findings

The findings were validated using an independent dataset or by lab experiments.

Questions for the team:
1. Were there any significant differences between patient characteristics of training and test set?
2. How were the 25 gene out of 243 chosen to establish the risk model?
3. Is there any evidence that this signature performs better than existing msigdb immune signatures? Especially T cell signature?

Reviewer 2 ·

Basic reporting

Nil.

Experimental design

Adequate.

Validity of the findings

Adequate.

Additional comments

In this study, the authors developed and validated a prognostic model based on the IRGs and clinical features to estimate OS in OC patients. They concluded that the IRG GBP1P1 might serve as prognostic biomarker for OC. The manuscript is straightforward, well written, and concise and has clear results. Definitely deserves to be published and is a valuable contribution to the “PeerJ” journal. However, the following comments need to be addressed, as recommended.

[1] “INTRODUCTION”, Lines 48-49:
“At present, ovarian cancer was the leading mortality of disease among gynecological cancers (Nash & Menon 2020; Siegel et al., 2023).”.
The term “ovarian cancer” should be specified. The authors should mention that 90% of ovarian cancers are of an epithelial cell type and comprise multiple histologic types, with various specific molecular changes, clinical behaviours, and treatment outcomes. The remaining 10% are non-epithelial ovarian cancers, which include mainly germ cell tumours, sex cord-stromal tumours, and some extremely rare tumours such as small cell carcinomas. Germ cell tumours differ to epithelial ovarian cancers with their earlier age of incidence, faster rate of growth, unilateral localisation (95% of cases) and good prognosis.
Recommended reference: Saani I, et al. Clinical Challenges in the Management of Malignant Ovarian Germ Cell Tumours. Int J Environ Res Public Health. 2023;20(12):6089.

[2] “INTRODUCTION”, Lines 50-52:
“The morbidity of OC was showing a rising trend because of advances in diagnostic technology and a growing national awareness of their health (Jemal et al., 2011; Nash & Menon 2020).”.
Moreover, the authors should mention that the number of asymptomatic ovarian masses has increased with the use of prenatal ultrasonography. Among ovarian tumours that complicate pregnancies, approximately 5% are malignant. Currently surgical intervention is indicated for an ovarian mass over 6 cm in diameter or when symptomatic.
Recommended reference: Boussios S, et al. A review on pregnancy complicated by ovarian epithelial and non-epithelial malignant tumors: Diagnostic and therapeutic perspectives. J Adv Res. 2018;12:1-9.

[3] “INTRODUCTION”, Lines 53-55:
“Therefore, there was an urgently need to find effective systemic therapy and identify novel biomarkers of prognosis assessment for OC.”.
At that point, the authors should clarify that currently, CA125 and HE4 are the only approved biomarkers for use in epithelial ovarian cancer. Importantly, miRNAs may have remarkable potential in various aspects of the prediction of the disease. However, further work is needed regarding its characterization as a biomarker. In particular, before miRNAs can be utilized as reliable biomarkers for clinical use, the steps involved in processing samples need to be standardized and the platforms for detecting miRNA in tumours and blood need to be refined.
Recommended reference: Ghose A, et al. Diagnostic biomarkers in ovarian cancer: advances beyond CA125 and HE4. Ther Adv Med Oncol. 2024 Feb 29;16:17588359241233225.

[4] “DISCUSSION”, Lines 363-364:
“The challenges majored in identification of the appropriate patients and finding effective combination therapeutic targets to amplify its clinical efficacy.”.
At that stage, the authors should also mention the therapeutic strategy of the combinations of PARP inhibitors with immunotherapies, such as anti-CTLA-4 and PD-1/PD-L1 that has partly been based on the hypothesis that BRCA1/2, and wild-type BRCA1/2 homologous recombination (HR) deficiency tumours display a higher neo-antigen load than HR-proficient cancers, producing more effective anti-tumour immune response. In addition, there is evidence that BRCA deficiency may induce a STING-dependent innate immune response, by inducing type I interferon and pro-inflammatory cytokine production. Interestingly enough, clinical models have also demonstrated that PARP inhibition inactivate GSK3 and upregulate PD-L1 in a dose-dependent manner. Consequently, T-cell activation is being suppressed, resulting in enhanced cancer cell apoptosis.
Recommended reference: Aliyuda F, et al. Advances in Ovarian Cancer Treatment Beyond PARP Inhibitors. Curr Cancer Drug Targets. 2023;23(6):433-446.

Reviewer 3 ·

Basic reporting

The author has clearly explained the methodology, and the paper flows nicely with effective graphical representations.

Experimental design

1. The GO and KEGG pathway analysis figures are based on q-values, while the text interprets results using p-values. Please ensure consistency in reporting the statistical values throughout the manuscript.

2. In line 379, it was mentioned that GBP1P1 is up-regulated on T-cells, it will be hard to conclude this just by looking at fetaurePlot, supporting evidence such as differential analysis has to be provided to verify these results.

3. It would be beneficial to include the results of the Univariate Cox regression analysis as a figure or as a table.

4. In fig s3 legend, GBP1P1 is mentioned, and in the figure, the axis label refers to LOC400759. Please keep the gene name consistent throughout.

5. In fig 3A legend, it says lasso regression of the 20 IRGs, I think it should be 25 IRGs.

Validity of the findings

The author did not cite the paper available at https://journals.sagepub.com/doi/10.1177/10732748231168756, which employs almost a similar approach to the current study. However, it should be noted that the present article includes some additional analyses and functional validation of the GBP1P1 gene. It would be beneficial if the author could review this paper and explain why it wasn't cited, and what aspects of their article enhance its contribution.

---

## Round 0.2 · accepted · Accept

Congratulations again.
Yours,
Yoshi
Prof. Yoshinori Marunaka, M.D., Ph.D.

Reviewer 1 ·

Basic reporting

I don't have any additional comments.

Experimental design

I don't have any additional comments.

Validity of the findings

I don't have any additional comments.

Additional comments

I don't have any additional comments.

Reviewer 2 ·

Basic reporting

The authors have successfully addressed my comments.
The revised manuscript should now be accepted for publication.
Thank you for inviting me to review this interesting manuscript.

Experimental design

The authors have successfully addressed my comments.
The revised manuscript should now be accepted for publication.
Thank you for inviting me to review this interesting manuscript.

Validity of the findings

The authors have successfully addressed my comments.
The revised manuscript should now be accepted for publication.
Thank you for inviting me to review this interesting manuscript.

Additional comments

The authors have successfully addressed my comments.
The revised manuscript should now be accepted for publication.
Thank you for inviting me to review this interesting manuscript.